# *Bacillus subtilis* RadA/Sms-Mediated Nascent Lagging-Strand Unwinding at Stalled or Reversed Forks Is a Two-Step Process: RadA/Sms Assists RecA Nucleation, and RecA Loads RadA/Sms

**DOI:** 10.3390/ijms24054536

**Published:** 2023-02-25

**Authors:** Rubén Torres, Begoña Carrasco, Juan C. Alonso

**Affiliations:** Department of Microbial Biotechnology, Centro Nacional de Biotecnología, CNB-CSIC, 28049 Madrid, Spain

**Keywords:** SsbA, RecO, RadA/Sms-RecA interaction, branch migration translocases

## Abstract

Replication fork rescue requires *Bacillus subtilis* RecA, its negative (SsbA) and positive (RecO) mediators, and fork-processing (RadA/Sms). To understand how they work to promote fork remodeling, reconstituted branched replication intermediates were used. We show that RadA/Sms (or its variant, RadA/Sms C13A) binds to the 5′-tail of a reversed fork with longer nascent lagging-strand and unwinds it in the 5′→3′ direction, but RecA and its mediators limit unwinding. RadA/Sms cannot unwind a reversed fork with a longer nascent leading-strand, or a gapped stalled fork, but RecA interacts with and activates unwinding. Here, the molecular mechanism by which RadA/Sms, in concert with RecA, in a two-step reaction, unwinds the nascent lagging-strand of reversed or stalled forks is unveiled. First, RadA/Sms, as a *mediator*, contributes to SsbA displacement from the forks and nucleates RecA onto single-stranded DNA. Then, RecA, as a *loader*, interacts with and recruits RadA/Sms onto the nascent lagging strand of these DNA substrates to unwind them. Within this process, RecA limits RadA/Sms self-assembly to control fork processing, and RadA/Sms prevents RecA from provoking unnecessary recombination.

## 1. Introduction

Replication fork progression is frequently impeded by physical barriers (damaged bases, DNA distortions, protein barriers), shortage of DNA precursors, or collision with transcription elongation complexes or R-loops [1,2,3,4]. This leads to replication stress, which is a major contributor to genomic instability [2,3]. In some cases, the lesion/obstacle is simply skipped. In other cases, homologous recombination (HR) functions may catalyze reversion of the stalled replication fork leading to a structure that resembles a Holliday junction (HJ), a process also called fork regression [2,3,5,6,7]. In both, the aim is to generate an intermediate for the resumption of DNA replication [2,3,5,6,7].

The model bacteria *Escherichia coli* and *Bacillus subtilis*, which diverged >1.5 billion years ago and show a genetic distance larger than that between humans and plants, use similar pathways to overcome a replication stress, albeit with different hierarchal order. (Unless stated otherwise, indicated genes and products were of *B. subtilis* origin.) In UV-irradiated *E. coli* cells, DNA synthesis drops within a few minutes, leading to a stable average number of replisomes [8] but a variable average number of DNA polymerase (DNAP) III*_Eco_* spots [8,9]. Here, RecA*_Eco_* forms foci at locations distal from replisomes in ~65% of the cases [8]. Similarly, RecO*_Eco_*, crucial for RecA loading, rarely coincide with DNAP III*_Eco_* [10]. It is likely that in *E. coli*, when a replication fork encounters a physical impediment, DNAP III*_Eco_* skips the barrier to resume synthesis downstream. Next, RecA*_Eco_* mainly assembles at the gap left behind and facilitates gap filling by different DNA damage tolerance sub-pathways [11,12,13]. However, up to ~35% of total RecA*_Eco_* foci colocalize with DNAP III*_Eco_* components, perhaps when lesion skipping is impeded by trafficking problems [4,14]. Here, DNAP III*_Eco_* disassembles, and the stalled fork is pushed backwards by an undefined branch migration translocase to anneal the nascent template strands, resulting in a reversed fork resembling a HJ structure [15,16]. Furthermore, RecBC*_Eco_* is the only recombination protein required for cell viability after replisome collision with transcription elongation complexes (or R-loops) [16].

Single cell analyses of unstressed exponentially growing *B. subtilis* cells reveal that there is only one active replisome in ~40% of cells [14]. During a single replication cycle, each fork requires restarting at least five times [14]. It is likely that: (i) the elongating replisomes are subjected to pervasive disassembly and reassembly events, an observation that is incompatible with the lesion skyping model, and (ii) the low abundant PriA, RecG, or RecO protein, which colocalizes at replication fork positions with ~2 foci/cell, travels with the replisome [17,18]. Independently of the nature of the barrier and whether the replisome disassembles or not, RecA forms foci that mainly colocalize with DNAP C subunits (85–96% colocalization) [19,20]. Concretely, in the absence of replisome disassembly, when the progression of either one or both replisome arms is impeded, RecA forms foci, which colocalize with DNAP C subunits (85% to 95% colocalization) but do not induce the SOS system [20,21,22]. In the presence of DNA lesions, the replisome disassembles, and SsbA coats the ssDNA gap and orchestrates the recruitment of several proteins [14]. One minute after UV-irradiation, RecA foci colocalize with the replisome in ~90% of cells. RecO, which is necessary for RecA foci formation, forms foci that colocalize with DNAP C subunits (>85% colocalization) [22]. As earlier proposed for mammalian cells [2,3,6,7], *B. subtilis*, in response to genotoxic insults, disengages the stalled replisome [14]. RecA assembled at a stalled fork protects it, facilitates fork remodeling by RecG or RadA/Sms, and controls fork reassembly by facilitating the recruitment of pre-primosome (DnaD) and replicative DNA helicase (DnaC) proteins [23,24,25,26,27].

RecA·ATP neither nucleates nor polymerizes onto SsbA-coated single-stranded DNA (ssDNA) [28,29]. The positive RecO mediator interacts with and antagonizes the negative effect of SsbA on RecA nucleation [29,30,31]. In fact, SsbA and RecO, as a two-component mediator, are necessary and sufficient to activate RecA to catalyze DNA strand exchange (DSE) [29]. Once homology is found, a RecA nucleoprotein filament catalyzes DNA strand invasion, and a displacement loop (D-loop) is formed. RecA, at the D-loop, physically interacts with and loads RadA/Sms [32,33,34]. Finally, RadA/Sms inhibits the ATPase of RecA and unwinds the D-loop [32,33,34] to facilitate double-strand break repair via extended synthesis-dependent strand annealing, as earlier proposed [35].

What is known about the ubiquitous bacterial RadA/Sms enzyme? RadA/Sms has four discrete domains: (i) a N-terminal C4-type zinc-binding motif, essential for the interaction with RecA (as observed when one of the cysteine residues of the motif is mutated to alanine, RadA/Sms C13A or C27A mutations) [32,34]; (ii) canonical RecA-like ATPase domains (defined by the RadA/Sms K104R or K104A mutations) [32,34]; (iii) a KNRFG motif essential for ATP hydrolysis (defined by the RadA/Sms K255R or K255A mutations) [32,34]; and (iv) a C-terminal P or LonC hexamerization domain similar to that of the *E. coli* hexameric ComM DNA helicase (absent in *B. subtilis*) and to the open spiral hexameric LonA protease [32,36,37]. Genetic and cytological analyses revealed that: (i) the *radA* gene is epistatic to *recA* in response to DNA damage [38,39]; (ii) in unperturbed exponentially growing wild type (*wt*) cells, RadA/Sms forms dynamic foci that mostly colocalize with the DNA bulk, while it forms static foci in the Δ*recG* context, where branched intermediates accumulate [38,40,41]; and (iii) the dominant negative effect of *radA* C13A, *radA* K104R, and *radA* K255R mutants on natural plasmid transformation is suppressed by *recA* inactivation [34,42].

Biochemical analyses revealed that at least Firmicutes and Deinococcus-Thermus RadA/Sms proteins adopt dumbbell-shaped homohexameric closed ring structures [32,36]. RadA/Sms has a central channel that accommodates ssDNA, a feature also observed in the C4 motif mutants (RadA/Sms C13A) [32,34]. *wt* or RadA/Sms C13A binds and unwinds DNA providing that there is an available 5′-tail, without the need of any accessory protein [32,34]. For instance, they bind to the 5′-tail of a replicating fork with a fully synthesized leading-strand and no-synthesis in the lagging-strand (3′-fork DNA) and unwind the template lagging-strand with similar efficiency [34]. However, they cannot unwind duplex or D-loop DNA substrates since they lack a 5′-tail [34]. It is likely that the 5′-tail of the template lagging-strand penetrates the central channel of the hexameric ring to unwind the 3′-fork DNA by moving in the 5′→3′ direction [32,34]. However, a DNA helicase activity for RadA*_Eco_* has not been described [43]. It has been proposed that RadA*_Eco_* speeds up the initial step of RecA*_Eco_*-mediated DNA strand exchange, as RecA mediators do [43,44].

RadA/Sms cannot unwind a replicating fork with a fully synthesized lagging-strand and no-synthesis in the leading-strand (5′-fork DNA). In the presence of RecA, *wt* RadA/Sms can unwind the nascent lagging-strand of 5′-fork DNA, but RadA/Sms C13A, which fails to interact with RecA, cannot [32,34,45]. In other words, RecA interacts with and loads RadA/Sms onto the nascent lagging-strand of 5′-fork DNA, and RadA/Sms unwinds it by a poorly understood mechanism [45]. However, many molecular details of the RecA activating mechanism for RadA/Sms unwinding, and of the role of RecA mediators on RadA/Sms substrate preference, are still unsolved. Moreover, very little is known of the RadA/Sms and RecA interplay in other bacteria.

In this study, we report findings that contribute to understanding the interplay of *B. subtilis* RadA/Sms and RecA in the presence of negative (SsbA) and positive (RecO) mediators. To unravel the role of these proteins, synthetic DNA substrates mimicking stalled replication forks with a small gap in the template leading- (forked-Lead) or lagging-strand (forked-Lag), or reversed forks with a nascent leading- (HJ-Lead) or lagging-strand tail (HJ-Lag) longer that its complementary strand, were constructed (Appendix A). We show that RadA/Sms assembles on the 5′-tail of HJ-Lag DNA and unwinds it, with RecA and its mediators (SsbA or RecO) limiting the unwinding reaction. A different outcome was observed when the HJ-Lead, forked-Lead, or forked-Lag DNA substrates were used. Here, RadA/Sms cannot unwind the substrate, and an interaction with RecA is necessary for unwinding. In the presence of SsbA, RadA/Sms, working as a specific *mediator*, antagonizes the negative effect exerted by SsbA on RecA nucleation onto the ssDNA 3′-tail of the HJ-Lead or ssDNA gap of forked-Lead or forked-Lag DNA. Here, RadA/Sms interacts with and inhibits the ATPase of RecA. Finally, RecA, as a specific *loader*, activates RadA/Sms to unwind their nascent lagging-strand, yielding (directly or indirectly) a 3′-fork product. This mechanism of replication fork restoration has two main advantages over canonical replication fork remodeling: a 3’-fork DNA intermediate cannot be cleaved by the RuvAB-RecU resolvasome and is a substrate suitable for PriA assembly [46].

## 2. Results

### 2.1. RadA/Sms Partially Counters the Negative Effect of SsbA on RecA Nucleation onto cssDNA

Previously, it has been shown that: (i) SsbA binds ssDNA with ~25-fold and >500-fold higher affinity than RadA/Sms·ATP and RecA·ATP, respectively [42,47]; (ii) SsbA blocks (by 20-fold, *p* < 0.01) the ATPase activity of RecA, competing with its nucleation and/or inhibiting its polymerization onto SsbA-coated ssDNA (Figure 1B, line 1 vs. 4, Appendix A) [28,29]; (iii) RadA/Sms and RadA/Sms C13A hydrolyze ATP with similar efficiency in the absence of cssDNA (Appendix A) [34]; (iv) RadA/Sms C13A-mediated ATP hydrolysis is stimulated by 5-fold (*p* < 0.01) in the presence of cssDNA, but that by *wt* RadA/Sms is not (Figure 1A,B, line 2, Appendix A) [34]; and (v) RadA/Sms interacts with and inhibits the ssDNA-dependent ATPase activity of RecA, but RadA/Sms C13A does not (Figure 1A,B, line 1 vs. 3, Appendix A) [34,42].

Then, we analyzed whether SsbA affects the ATPase activity of RadA/Sms and if SsbA and RadA/Sms additionally affect RecA filament growth. SsbA cannot hydrolyze ATP [47]. In the presence of cssDNA, the ATPase activity of RadA/Sms (1 RadA/Sms hexamer/330-nt) was not apparently affected by SsbA (1 SsbA tetramer/66-nt) (Figure 1A, line 2 vs. 5, Appendix A).

When RecA (1 RecA monomer/12.5-nt), SsbA, and RadA/Sms were incubated with cssDNA, the maximum efficiency of ATP hydrolysis was increased by ~10-fold (*p* < 0.01) when compared to the one observed by the RecA-cssDNA-SsbA complex (Figure 1A, line 1 vs. 4 and 6, Appendix A). The maximal rate of ATP hydrolysis, however, was still decreased by ~2-fold (*p* < 0.01) when compared with RecA alone (Figure 1A, line 1 vs. 6, Appendix A). It is likely that RadA/Sms partially counteracts the negative effect of SsbA on RecA ATPase activity. This effect cannot be attributed to a protein–protein interaction, because neither RadA/Sms (Appendix A) nor RecA [47] physically interacts with SsbA.

To further evaluate if RadA/Sms can displace SsbA from ssDNA, the *wt* protein was replaced by RadA/Sms C13A, since: (i) its ATP hydrolysis is stimulated in the presence of cssDNA, and (ii) it fails to interact with RecA and to inhibit is ATPase activity [34]. SsbA reduced by ~2.5-fold the ATPase activity of RadA/Sms C13A (*p* < 0.01) (Figure 1B, line 2 vs. 5, Appendix A). The maximal efficiency of ATP hydrolysis in the presence of RadA/Sms C13A, RecA, SsbA, and cssDNA was significantly higher (~15-fold, *p* < 0.01) when compared with the maximal rate of ATP hydrolysis by the RecA-cssDNA-SsbA complex (Figure 1B, line 6 vs. 4, Appendix A). It is likely that the *wt* and RadA/Sms C13A enzymes can partially counter the negative effect of SsbA on the maximal efficiency of ATP hydrolysis by RecA. Here, the ssDNA-independent ATPase activity of RadA/Sms C13A may mask the analysis.

### 2.2. The RecO-SsbA Mediator Poorly Counteracts the Negative Effect of RadA/Sms on RecA ATPase

Previously, it has been shown that: (i) RecO cannot hydrolyze ATP [30], and (ii) RecO (1 RecO monomer/50-nt) partially displaces SsbA and facilitates RecA nucleation and filament growth onto ssDNA coated by saturating SsbA concentrations (1 SsbA tetramer/33-nt) [29,30]. Thus, we analyzed whether SsbA and RecO counteract the negative effect that RadA/Sms exerts on the ATPase of RecA.

First, we confirmed that both the ssDNA-dependent or ssDNA-independent ATPase activity of *wt* RadA/Sms or RadA/Sms C13A was not affected by the addition of a *circa* saturating RecO concentration (1 RecO monomer/100-nt) (Appendix A). To avoid the noise introduced by the ssDNA-independent ATPase activity of *wt* RadA/Sms, it was replaced by the ATPase defective RadA/Sms K104R or RadA/Sms K255R mutant variants that still physically interact with RecA [34]. These mutant variants (1 hexamer/330-nt) also blocked the ATPase activity of RecA (*p* < 0.01) (Figure 1C,D, line 1 vs. 2, Appendix A). This inhibition is protein specific, because the ATPase activity of an unrelated ssDNA-dependent ATPase (PcrA) was not affected by *wt* RadA/Sms, RadA/Sms K104R, or RadA/Sms K255R (Appendix A).

RecO (1 RecO monomer/100-nt) was sufficient to partially displace SsbA (1 SsbA tetramer/66-nt) and significantly increased the maximal rate of ATP hydrolysis by RecA (1 RecA monomer/12.5-nt) (Figure 1A line 4 vs. Figure 1C,D, line 3, Appendix A). However, a preformed RecO-cssDNA-SsbA complex was unable to antagonize the negative effect of RadA/Sms K104R or RadA/Sms K255R on the ATPase activity of RecA (*p* < 0.01) (Figure 1C,D, line 2 vs. 4). Similar results were observed when the two-component mediator (SsbA-RecO) was added to pre-formed RecA-cssDNA-RadA/Sms K104R (or RecA-cssDNA-RadA/Sms K255R) complexes (Figure 1C,D, line 2 vs. 5). Nevertheless, when RadA/Sms K104R or RadA/Sms K255R was added to preformed RecO-SsbA-cssDNA-RecA complexes, RecA-mediated ATP hydrolysis was recovered (*p* < 0.01) when compared with RecA-ssDNA-RadA/Sms complexes (Figure 1C,D, line 2 vs. 6 and Appendix A).

From these data altogether we conclude that: (i) RadA/Sms-RecA, RadA/Sms K104R-RecA or RadA/Sms K255R-RecA complexes limit the ATPase of RecA and indirectly RecA dynamics; (ii) the presence of the two-component mediator (SsbA and RecO) cannot counteract RadA/Sms K104R or RadA/Sms K255R (Appendix A); (iii) the interplay between RecA with RadA/Sms or SsbA-RecO may be mutually exclusive, because RecA nucleated on the preformed SsbA-cssDNA-RecO complexes is only partially sensitive to RadA/Sms K104R- or RadA/Sms K255R-mediated inhibition (Figure 1C,D, line 6); and (iv) RadA/Sms interacts with and loads RecA on the SsbA-cssDNA complexes, but inhibits RecA redistribution on cssDNA (Appendix A).

To further test whether RadA/Sms or its mutant variants (RadA/Sms C13A (unable to interact with RecA) or RadA/Sms K104A (unable to hydrolyze ATP)) can affect the enzymatic activity of RecA, a three-strand recombination assay was used. As revealed in Appendix A, RecA in the ATP bound form (RecA·ATP) or RecA·dATP-mediated DSE was significantly reduced and inhibited (*p* < 0.01) in the presence of ~6 and ~12 RadA/Sms, RadA/Sms C13A, or RadA/Sms K104A hexamers/cssDNA molecule, respectively (Appendix A). Alternatively, RadA/Sms bound to the DNA substrates reverses RecA·ATP-mediated DSE. Independently of the time point at which RadA/Sms is added, the reaction neither progressed further nor is reversed when compared to the absence of RadA/Sms (Appendix A). It is likely that six or more RadA/Sms, RadA/Sms C13A, or RadA/Sms K104A hexamers/cssDNA molecule limit RecA-driven DSE rather than stimulating the backwards reaction toward the original substrates. An uncharacterized function(s) should be necessary to overcome RadA/Sms-mediated limitation of DSE, because RadA/Sms is crucial for strand assimilation during natural chromosomal transformation [32,34].

### 2.3. RadA/Sms Catalyzes Unwinding of Reversed Forks with Longer Nascent Lagging-Strand, but RecA and SsbA Reduce Unwinding

RecA forms foci that colocalize with the stalled replisome in >90% of cells after UV-irradiation [19,22]. RadA/Sms forms dynamic foci that mostly colocalize with the DNA bulk in *wt* cells, but it forms static foci in the Δ*recG* context, where branched intermediates accumulate [38,40,41]. In vitro, SsbA bound to ssDNA blocks RecA nucleation on the SsbA-ssDNA complex, but RadA/Sms partially displaces SsbA from ssDNA (Figure 1). Furthermore, RecA binds and protects branched intermediates (e.g., stalled or reversed fork) and interacts with and loads RadA/Sms at these substrates [34]. To test whether RadA/Sms in the presence of RecA and/or SsbA remodels or processes a stalled fork with a leading-strand gap that is reversed, a synthetic reversed fork with a nascent lagging-strand longer than the leading-strand (HJ-Lag) was constructed (Appendix A).

The size site of RadA/Sms is unknown. To design the HJ-Lag substrate, we first analyzed the efficiency of unwinding of duplex DNAs with increasing 5′-tail lengths (Appendix A). RadA/Sms poorly unwound a 15-nt long 5′-tail duplex, but it efficiently unwound a substrate with a 30-nt long 5′-tail (Appendix A). It is likely that the RadA/Sms site size lay between these two values. The size sites of SsbA and RecA were also analyzed. SsbA binds ssDNA with a ~30-nt average size (among the SSB_65_, SSB_35_, SSB_15_, and SSB_7_ binding modes) [47,48]. RecA·ATP forms a discrete dynamic RecA nucleoprotein filament when ~8 or more monomers bind ssDNA, with one monomer binding 3-nt [49,50,51]. Thus, a HJ-Lag DNA substrate with a nascent lagging-strand 30-nt longer than the nascent leading-strand was constructed (Appendix A).

RadA/Sms·ATP or RadA/Sms C13A·ATP (10 nM) bound and unwound the nascent lagging-strand of the HJ-Lag DNA substrate with similar efficiency (Figure 2A,B, lane 4). Since forked DNA products that would be generated by coupling DNA unwinding to duplex rewinding were not observed, we discarded a branch migration activity to restore a replication fork, as observed with the RecG or RuvAB branch migrating translocases [26,33,48,52]. Instead, RadA/Sms or RadA/Sms C13A, in a first step, unwound the nascent lagging-strand. When the protein concentration was increased (e.g., to 20 nM), the 3-way junction intermediate was further processed to yield a 5′-tailed flapped intermediate and, finally, the labelled strand product (Figure 2A,B, lane 5, and S6A). It is likely that: (i) the 5′-tail of a HJ-Lag DNA penetrates the central hole of a RadA/Sms or RadA/Sms C13A hexamer; and (ii) RadA/Sms unwinds the nascent lagging-strand to yield, upon spontaneous branch migration, a suitable substrate for replication restart: a 3′-fork DNA (Figure 2A,B).

Saturating RecA or SsbA, with respect to ssDNA, inhibited ~50% the unwinding of the nascent lagging-strand and blocked the accumulation of the radiolabeled nascent leading-strand product (Figure 2A, lane 5 vs. lanes 12 and 16). Similar results were observed when *wt* RadA/Sms was replaced by RadA/Sms C13A (Figure 2B). The inhibition is independent of a protein–protein interaction because SsbA does not interact with RadA/Sms (Appendix A) and RecA does not interact with RadA/Sms C13A [34]. It is likely that RecA or SsbA partially outcompetes RadA/Sms or RadA/Sms C13A for assembling on HJ-Lag DNA and subsequent intermediates (Appendix A).

To test the hypothesis, the combined action of RecA and SsbA over RadA/Sms (or RadA/Sms C13A) was analyzed by changing the order of protein addition (Figure 2B,C). First, the HJ-Lag DNA was pre-incubated with fixed RadA/Sms (or RadA/Sms C13A) and SsbA concentrations, and then 2 mM ATP and increasing RecA concentrations were added (Figure 2B,C, lanes 9–11). At a RadA/Sms (RadA/Sms C13A):RecA:SsbA 1:5:15 molar ratio, unwinding was reduced by ~2-fold, but at a 1:20:15 ratio, unwinding was significantly reduced (~4.5-fold), and only the nascent lagging-strand was unwound (Figure 2B,C, lane 11). When preformed RadA/Sms-HJ-Lag (or RadA/Sms C13A-HJ-Lag) complexes were incubated with fixed SsbA and increasing RecA concentrations and ATP, DNA unwinding was not significantly affected (*p* > 0.1) (Figure 2B,C, lanes 12–14 vs. 5). It is likely that neither RecA nor SsbA displaces *wt* RadA/Sms (or RadA/Sms C13A) once it is loaded on the 5′-tailed HJ-Lag DNA.

To further examine the limitation on RadA/Sms unwinding of HJ-Lag DNA by SsbA or RecA, a kinetic analysis of DNA unwinding, varying the order of protein addition, at a RadA/Sms:RecA:SsbA 1:10:15 ratio, was undertaken (Figure 2D). In a 40 min reaction, the reversed fork was fully unwound by RadA/Sms, mainly rendering the labelled nascent leading-strand, but RecA addition reduced the unwinding reaction by ~3.5-fold (*p* < 0.01) (Figure 2D, lane 3 vs. 4). When the DNA substrate was pre-incubated with RadA/Sms and SsbA, and then RecA and ATP were added, at min 10, only the nascent lagging-strand was unzipped (Figure 2D, lane 5). Then, the template lagging-strand (min 15), and finally, the flayed intermediate (min 20) were unwound to render the labelled nascent leading-strand product (Figure 2D, lanes 6–9 and Appendix A). When RadA/Sms was pre-incubated with the substrate, and then fixed RecA, SsbA, and ATP were added, the flayed intermediate and the final product were observed at earlier times (Figure 2D, lanes 10–14).

From these data, it is likely that: (i) SsbA and RecA polymerizing toward the junction reduce RadA/Sms loading on the 5′-tail of HJ-Lag DNA, and additionally inhibit RadA/Sms unwinding of the nascent lagging-strand and the intermediates; and (ii) RecA and SsbA cannot outcompete a preformed RadA/Sms-HJ-Lag DNA complex.

### 2.4. How RecA Activates RadA/Sms to Unwind a HJ Structure with a Nascent Leading-Strand Longer Than the Nascent Lagging-Strand

When a replication fork stalls due to a lesion on the lagging-strand, it may form a HJ-like structure with a nascent leading-strand longer than the nascent lagging-strand (HJ-Lead DNA) upon reversal (Appendix A). It has been shown that a HJ-Lead structure, which lacks an available 5′-tail, cannot be unwound by RadA/Sms. However, RecA activates RadA/Sms to unwind its nascent lagging-strand by a poorly understood mechanism [45].

Eight different scenarios (*a* to *g*) can be envisioned to explain how RadA/Sms in concert with RecA unwinds the nascent lagging-strand of HJ-Lead DNA (Appendix A): (*a*) traces of a contaminating 3′→5′ ssDNA exonuclease in the RecA preparation may degrade the 3′tail of the HJ-Lead DNA to expose the 5′-tailed strand, with subsequent RadA/Sms self-loading; (*b*) dynamic RecA may partially open the duplex DNA at the junction to generate a RadA/Sms entry site (Appendix A); (*c*) RadA/Sms may activate RecA to remodel the HJ substrate leading to flapped intermediates; (*d*) RecA bound to the 3′-tailed nascent leading-strand of HJ-Lead DNA might fray the nascent lagging-strand to promote RadA/Sms self-loading to the end of the nascent lagging-strand (Appendix A); (*e*) in vivo RadA/Sms may be in a monomer–hexamer equilibrium in solution, and RecA may interact with and construct a ring-shaped RadA/Sms hexamer around the nascent lagging-strand from monomeric subunits (ring-maker strategy); (*f*) in the absence of ssDNA, RadA/Sms may adopt an open-spiral hexameric structure, as reported for LonA*_Eco_* in the absence of a substrate [37], and RecA nucleated onto the ssDNA might interact with, load, and close the RadA/Sms open hexameric ring around the nascent lagging-strand (ring-closure strategy); or (*g*) a hexameric RecA right-handed ring [53] with a 1:1 stoichiometric relative to RadA/Sms may physically open a preformed RadA/Sms hexameric ring and load it onto the nascent lagging-strand (ring-breaker strategy), as reported for the replicative DnaB*_Eco_* helicase or the β sliding clamp [54,55]. As stated in Appendix A, we have considered unlikely conditions (*a*) to (*f*) and favored the ring-breaker hypothesis (condition *g*), unveiling a novel role of RecA. The sequence of the events to activate unwinding of the nascent lagging-strand—RecA complexed with hexameric RadA/Sms before interacting with the SsbA-ssDNA complex or RecA bound to ssDNA recruiting RadA/Sms on the nascent lagging-strand—is still unclear.

### 2.5. Two Step RadA/Sms Loading on the Nascent-Lagging Strand of HJ-Lead DNA

It has been shown that RecA·ATP neither nucleates nor polymerizes on the SsbA-ssDNA complexes [34,42]. Then, SsbA bound to the 3′-tail of the HJ-Lead DNA should indirectly inhibit RecA activation of RadA/Sms to unwind the nascent-lagging strand. Nevertheless, in the presence of RadA/Sms or RadA/Sms C13A, SsbA can be partially displaced and the ATPase activity of RecA partially recovered (Figure 1). To test whether RadA/Sms can antagonize the negative effect of SsbA on RecA binding to the 3′-tail of the HJ-Lead DNA and unwind it, a synthetic substrate with a nascent leading-strand 30-nt longer than the nascent lagging-strand was constructed (Figure 3 and Appendix A).

RadA/Sms, SsbA, or RecA cannot unwind the HJ-Lead DNA, but RecA activates RadA/Sms to unwind its nascent lagging strand (Figure 3A, lines 3–5 vs. 6) [45]. A preformed SsbA-HJ-Lead DNA complex, at a RadA/Sms:RecA:SsbA 1:5:15 molar ratio, inhibited RadA/Sms unwinding of the HJ-Lead DNA (*p* < 0.01) (Figure 3A, lane 9 vs. 6). The presence of increasing RecA, to a RadA/Sms:RecA:SsbA 1:10:15 molar ratio, was necessary and sufficient to assist RadA/Sms-mediated unwinding of the nascent lagging-strand, yielding a 3-way DNA intermediate (Figure 3A, lane 10 and Appendix A). A further increase in RecA, to a 1:20:15 molar ratio, stimulated RadA/Sms-mediated unwinding to levels comparable to the absence of SsbA (Figure 3A, lane 11 vs. 6). It is likely that: (i) sub-stoichiometric RecA concentrations relative to SsbA (~0.6-fold) are sufficient for RadA/Sms unwinding of the nascent-lagging strand; (ii) RecA cannot nucleate onto SsbA-coated ssDNA even in the presence of a ~2.7-fold excess relative to SsbA in the absence of RadA/Sms (Figure 1); and (iii) RadA/Sms, in concert with RecA, displaces SsbA and facilitates RecA loading on the 3′-tailed HJ-Lead DNA, polymerizing away from the junction.

To test whether SsbA competes with preformed RadA/Sms-HJ-Lead-RecA complexes, fixed RadA/Sms and increasing RecA concentrations were pre-incubated with the DNA substrate, and then a saturating SsbA concentration and 2 mM ATP were added. After 15 min incubation, the unwinding reaction was not significantly affected, reaching levels comparable to those in the absence of SsbA (*p* > 0.1) (Figure 3A, lanes 12–14 vs. 6–8). This suggests that: (i) SsbA cannot outcompete the preformed RadA/Sms-HJ-Lead-RecA complexes (Figure 3A, lanes 12–14); and (ii) in the presence of RadA/Sms, a RecA filament, at a RadA/Sms:RecA:SsbA 1:10:15 molar ratio, is insensitive to SsbA competition.

To analyze whether RadA/Sms antagonizes SsbA and facilitates RecA loading on the 3′-tailed HJ-Lead DNA, the DNA substrate was pre-incubated with fixed RadA/Sms and SsbA concentrations, and then ATP and increasing RecA were added. At a RadA/Sms:RecA:SsbA 1:5:15 molar ratio, RadA/Sms moderately catalyzed unwinding of the nascent lagging-strand (Figure 3A, lane 15). However, at a RadA/Sms:RecA:SsbA 1:10:15 molar ratio, the RadA/Sms unwinding efficiency was comparable to the assay in the absence of SsbA (Figure 3A, lanes 16–17). This suggests that RadA/Sms may work as a specialized *mediator* provided that > 5 RecA monomers/RadA/Sms hexamer are present in the reaction.

To follow how DNA unwinding occurs, a kinetic analysis was performed with fixed RadA/Sms, RecA, and SsbA concentrations at a 1:10:15 RadA/Sms:RecA:SsbA molar ratio (Figure 3B). RadA/Sms, in concert with RecA, unwound ~95% of the HJ-Lead DNA substrate in a 40 min reaction (Figure 3B, lane 3). When the substrate was pre-incubated with SsbA, and then RadA/Sms, RecA, and ATP were added, at 10 min, only the nascent lagging-strand was unwound, leading to a 3-way junction (Figure 3B, lane 4). With a longer incubation time, the 5′-tailed flapped intermediate also accumulated (min 15), and at later times (min 40), the labelled nascent leading-strand was observed (Figure 3B, lanes 5–8, Appendix A). When the DNA substrate was pre-incubated with RadA/Sms and RecA, and then SsbA and ATP were added, at earlier times (min 10), the nascent and the template lagging-strands were unwound, and at later times (min 20), the labelled leading-strand was also unwound (Figure 3B, lanes 9–13). In summary, the final product was formed with higher efficiency and earlier under these conditions than when SsbA was pre-incubated with the DNA (Figure 3B, lanes 9–13 vs. 4–8).

The requirement of a RadA/Sms-RecA interaction was confirmed by replacing *wt* RadA/Sms by RadA/Sms C13A, a variant that cannot interact with RecA [34]. Independently of the order of protein addition, the RadA/Sms C13A variant did not unwind the HJ-Lead DNA substrate (Figure 3C, lanes 6–17).

Altogether, these data suggest that RadA/Sms-mediated unwinding of a HJ-Lead DNA is a two-step process. First, RadA/Sms counteracts the negative effect of SsbA on RecA loading onto ssDNA and facilitates RecA nucleation and filament growth on the ssDNA away from the junction. However, upon RadA/Sms-RecA interaction, RecA dynamics is inhibited, as judged by RecA-mediated ATP hydrolysis (Figure 1). Second, interaction of paused RecA·ATP or RecA·ATPγS filamented on the nascent leading-strand of HJ-Lead DNA is necessary and sufficient to load RadA/Sms onto the nascent lagging-strand, at above a RadA/Sms:RecA 1:5 stoichiometry, and unwind it from the HJ-Lead DNA substrate (Figure 3A, Appendix A). To confirm that the formation of a RecA filament onto the ssDNA (and not RecA in solution) is necessary for RadA/Sms activation, an artificial substrate (a replicating fork with fully synthesized leading- and lagging-strands (3′–5′-fork DNA)) was constructed (Appendix A). As revealed in Appendix A, RecA cannot activate RadA/Sms to remodel this DNA substrate, suggesting that RecA bound to ssDNA is necessary for RadA/Sms unwinding of the nascent lagging-strand.

### 2.6. RecA Mediators Increase RecA Dynamics and Limit RadA/Sms Unwinding of Reversed Forks

RecO interacts with and partially displaces SsbA from ssDNA, and both proteins stimulate RecA redistribution on the ssDNA (Figure 1C,D, line 3) [29]. In previous sections, it was shown that: (i) RadA/Sms interacts with and inhibits RecA dynamics (Figure 1); (ii) RecA and SsbA outcompete RadA/Sms (or RadA/Sms C13A) for binding onto HJ-Lag DNA, and thereby reduce unwinding of the nascent lagging-strand (Figure 2); (iii) RecA, filamented on the 3′tail of a HJ-Lead DNA substrate, interacts with, loads, and activates RadA/Sms to unwind the nascent lagging-strand; and (iv) depending on the protein stoichiometry and the order of protein addition, SsbA may inhibit the unwinding reaction (Figure 3). To analyze whether the two-component mediator (RecO and SsbA), by increasing RecA dynamics, affects RadA/Sms binding and unzipping of HJ-Lag DNA, kinetic analyses of RadA/Sms-mediated unwinding, varying the order of protein addition, were undertaken (Figure 4).

In a 40 min reaction, RadA/Sms bound to the 5′-tail of HJ-Lag DNA fully unwound the DNA substrate, but SsbA (Figure 2A, lanes 14–16), RecO, or RecA reduced the unzipping reaction (Figure 4A, lane 4 vs. 5–6). When SsbA, RecO, and RadA/Sms were pre-incubated with HJ-Lag DNA, and then RecA and ATP were added, RadA/Sms did not unwind the substrate during the first 15 min of incubation (Figure 4A, lanes 7–8). After 20 min of incubation, only a 3-way junction intermediate was observed (Figure 4A, lane 9). It is likely that the unwinding reaction was reduced and delayed when compared with the condition in which RecO was absent (Figure 2D, lanes 5–7). Conversely, if RadA/Sms was pre-incubated with HJ-Lag DNA, and then SsbA, RecO, RecA, and ATP were added, the reaction occurred faster, but not to the level reached when RecO was omitted (Figure 4A, lanes 12–16 vs. Figure 2D, lanes 10–14). It is likely that dynamic RecA, by the action of RecO-SsbA, outcompetes RadA/Sms recruitment on 5′-tailed HJ-Lag DNA more efficiently than RecA alone.

Then, we analyzed whether RecO and SsbA affect RecA activation of RadA/Sms-mediated unwinding of the HJ-Lead DNA (Figure 4B). RecO did not activate RadA/Sms to unwind this substrate (Figure 4B, lane 4). When the substrate was pre-incubated with SsbA and RecO, and then RadA/Sms, RecA, and ATP were added, the unwinding of the nascent and of the template lagging-strand was comparable to that when RecO was omitted (Figure 4B, lanes 6–10 vs. Figure 3B, lanes 4–8). This confirms that RadA/Sms counteracts RecO-facilitated RecA dynamics (see Figure 1C,D). The preformed RadA/Sms-HJ-Lead DNA-RecA complexes unwound the nascent lagging-strand more efficiently than when SsbA and RecO were pre-bound to the DNA substrate (Figure 4B, lanes 6–10 vs. 11–15). It is likely that: (i) RecO and SsbA limit RadA/Sms’s unwinding of the nascent lagging-strand of the HJ-Lead DNA substrate, and (ii) RadA/Sms’s interaction with RecA polymerizing away from the junction overrides RecA dynamics enhanced by RecO-SsbA.

### 2.7. RecA Activates RadA/Sms Unwinding of the Nascent Lagging-Strand of Synthetic Stalled Forks

In previous sections, it was shown that: (i) RadA/Sms binds the 5′-tail of HJ-Lag DNA and unwinds the substrate, with RecA and its accessory proteins outcompeting RadA/Sms for binding to the DNA substrate (Figure 2, Figure 4A and Appendix A); and (ii) RadA/Sms, in concert with RecA, at a 1:10 molar ratio, can overcome the negative effect of SsbA, or of SsbA and RecO, on the unwinding of the nascent lagging-strand of HJ-Lead DNA substrate (Figure 1, Figure 3, and Appendix A). DNA structures mimicking stalled replication forks caused by blockage on the leading- or lagging-strand (Appendix A) were designed to: (i) gain insight into how RadA/Sms processes branched intermediates; (ii) unravel the contribution of RadA/Sms as a mediator, pausing RecA; and (iii) understand the contribution of RecA as a specialized loader.

RecA nucleated on a 15-nt long ssDNA does not hydrolyze ATP, enters into a “paused” state [50,56], and RadA/Sms poorly interacts with a 15-nt long ssDNA (Appendix A). Then, to mimic a paused RecA and to reduce the probability of RadA/Sms self-recruitment, the length of the gapped ssDNA in the stalled fork substrate was decreased to 15-nt, either in the parental lagging- (forked-Lag) or leading-strand (forked-Lead) (Appendix A). To confirm the integrity of the substrates, they were incubated with the branch migrating translocase RecG, which remodels a stalled fork with a leading- or a lagging-strand gap to yield a reversed fork [48]. In our assay, the nascent strands are not complementary, hence, the product should be an unreplicated fork [33]. As expected, the RecG (15 nM) control (C) unwound ~50% of nascent strands to render an unreplicated fork product with similar efficiency in the two substrates (Figure 5A,B, lane 18).

RecA, SsbA, or RadA/Sms neither reversed the forked-Lag or forked-Lead DNA substrate nor unwound their nascent strands (Figure 5A,B, lanes 3–5). In the presence of RecA, however, RadA/Sms unwound the nascent lagging-strand of both substrates with similar efficiency (Figure 5A,B, lanes 6–8). When *wt* RadA/Sms is replaced by RadA/Sms C13A, no unwinding was observed. It is likely that: (i) RecA nucleated toward the junction on the gapped forked-Lag DNA or away from the junction on the gapped forked-Lead DNA is necessary and sufficient to activate RadA/Sms to unwind the nascent lagging-strand of both DNA substrates, and (ii) RecA interacts with and modulates the conformation of hexameric RadA/Sms and loads it in *cis* or in *trans* on the nascent lagging-strand (Appendix A).

To analyze whether SsbA binds to and inhibits RecA nucleation onto the ssDNA gap and indirectly blocks RecA activation of RadA/Sms-mediated unwinding of these stalled fork substrates, and if RadA/Sms alone or in concert with RecA antagonizes the negative effect of SsbA on RecA nucleation, the order of protein addition was varied. When a preformed SsbA-forked-Lag or SsbA-forked-Lead complex was incubated with RadA/Sms and RecA (at a RadA/Sms:RecA:SsbA 1:5:15 stoichiometry), RadA/Sms-mediated unzipping of the nascent lagging-strand was not observed (Figure 5A,B, lane 9). Increasing RecA concentrations, to RadA/Sms:RecA:SsbA 1:10:15 and 1:20:15 molar ratios, were necessary to counteract the negative effect of SsbA on RadA/Sms unwinding of the nascent lagging-strand of both DNA substrates (Figure 5A,B, lanes 10–11). It is likely that a RecA-RadA/Sms stoichiometry of 1:10 or 1:20 is necessary and sufficient to antagonize the negative effect of SsbA on RecA nucleation and subsequent loading of RadA/Sms.

When SsbA was added to preformed RecA-forked-Lag-RadA/Sms or RecA-forked-Lead-RadA/Sms complexes, the rate of RadA/Sms-mediated unwinding was similar to that in the absence of SsbA (Figure 5A,B, lanes 6–8 and 12–14). This suggests that SsbA cannot act against RadA/Sms-RecA bound to forked-Lag or forked-Lead DNA. Finally, when SsbA, RadA/Sms, and forked-Lag or forked-Lead DNA were pre-incubated, and then fixed ATP and increasing RecA concentrations were added, RadA/Sms-mediated unwinding of the nascent lagging-strand was not significantly different when compared with the condition of RadA/Sms and RecA first (Figure 5A,B, lanes 15–17 vs. 12–14). It is likely that RadA/Sms facilitates the displacement of SsbA, with RecA being necessary to load RadA/Sms on the nascent lagging-strand.

From the data presented in Figure 5, we can predict (Appendix A) that: (i) RecA nucleates on a 15-nt gap, but it cannot remodel the small synthetic stalled forks; (ii) RadA/Sms neither remodels nor unwinds the synthetic forked-Lag or forked-Lead DNA substrate; (iii) the 5′-end of nucleated RecA on a 15-nt long ssDNA gap is necessary and sufficient to direct RadA/Sms, both *in cis* or *in trans*, to their nascent lagging-strand; (iv) saturating SsbA bound to gapped ssDNA, perhaps in the SsbA_15_ binding mode, outcompetes RadA/Sms unwinding at a RadA/Sms:RecA:SsbA 1:5:15 molar ratio; (v) RadA/Sms, in concert with RecA, as a mediator, interacts with and facilitates RecA loading on the SsbA-ssDNA gapped DNA substrate; and (vi) about 6 RecA monomers in solution may open a hexameric ring and, upon interaction with 5 RecA nucleated at the 15-nt long gapped forked-Lag or forked-Lead DNA, assemble RadA/Sms on the nascent lagging-strand.

## 3. Discussion

*B. subtilis* respond to a replicative stress by forming RecA foci, in concert with SsbA and RecO, that colocalize with the replisome (colocalization > 90%) at a stalled replication fork [19,22]. RadA/Sms forms dynamic foci in *wt* cells but static foci in Δ*recG* cells, a mutation that causes accumulation of branched intermediates [38,57]. In this study, we sought to decipher whether paused RecA and ring-shaped homohexameric RadA/Sms define a novel function upon replication stress to promote replication restart. In summary, first, RecA interacts with and loads RadA/Sms at stalled forks [34,45]. Second, RadA/Sms inhibits RecA-mediated ATP hydrolysis, crucial for its dynamic activity (Figure 1). Third, RadA/Sms loaded at the nascent lagging-strand of stalled or reversed forks unwinds it to generate, directly or indirectly, a 3′-fork DNA substrate (Figure 3 and Figure 5). Fourth, the pre-primosome PriA-DnaD-DnaB proteins recognize the 3′-fork DNA, and with DnaI, load the replicative DnaC helicase for replication restart [46]. In fact, in the absence of RecA, recruitment of pre-preprimosome DnaD and replicative DNA helicase DnaC is significantly reduced [23].

From previous studies and the data presented here, six ideas can be drawn. First, cytological analyses have shown that SsbA, when travelling with the replisome, may load RecO, RecG, or PriA at a stalled replication fork [17,18]. RecO contributes to RecA foci formation at stalled replication forks [19,22], RecG is a genuine fork remodeler, and PriA is required at a later stage for replication re-initiation [46,48].

Second, SsbA interacts with and loads RecO onto a lesion-containing gap, then RecO partially displaces SsbA and generates the substrate for RecA nucleation [29]. Here, RecA, in concert with its mediators and modulators, would catalyze DNA invasion and strand transfer, crucial for template switching and double-strand break repair [58]. Alternatively, RadA/Sms interacts with and limits RecA-mediated ATP hydrolysis (Figure 1) and strand transfer (Appendix A). This way, RadA/Sms may prevent RecA from provoking unnecessary DNA strand exchange at stalled or reversed forks.

Third, RecA neither nucleates nor polymerizes on the SsbA-ssDNA complexes [28,29,30]. However, RadA/Sms, as a specialized mediator, stimulates RecA nucleation onto ssDNA. Concretely, RadA/Sms counteracts the negative effect of SsbA on RecA nucleation on the 30-nt long 3′-tail of a HJ-Lead DNA substrate, or on the 15-nt gap at the leading or lagging template strand of a forked-Lead or forked-Lag DNA, and limits RecA redistribution, even in the presence of the positive RecO mediator (Figure 1, Figure 3, and Figure 6A,B). It is unclear whether RadA/Sms alone or upon a transient interaction with RecA undergoes a structural change to displace SsbA and load RecA. At present, we cannot rule out that RadA/Sms modulates RecA conformation to a form that can displace SsbA from ssDNA, although we favor the former hypothesis.

Fourth, RecA·ATP (or RecA·ATPγS) protects the nascent strands from degradation and loads RadA/Sms on the nascent lagging-strand of stalled forks (or of reversed forks with a nascent leading-strand longer than the nascent lagging-strand) by a poorly understood mechanism (Figure 6A,B,D). RecA·ATP forms right-handed helical filaments of 6.2 RecA protomers per helix turn on ssDNA, extending it by 50% relative to B-form dsDNA [51,53]. If each RecA monomer interacts with each protomer of RadA/Sms, the hexameric form of the latter may open and ssDNA may pass through the gap in the ring. Then, RecA would deliver RadA/Sms in *cis* or in *trans* onto the nascent lagging-strand to render a 3′-fork DNA specific for replication restart. This mechanism mimics loading of the replicative helicase DnaB*_Eco_* by DnaC*_Eco_* self-oligomerized as a right-handed helical filament by a ring-breaking strategy [54].

Fifth, activated RadA/Sms unwinds the nascent lagging-strand of stalled forks (Figure 6A,B) or of reversed forks (HJ-Lead DNA) (Figure 6D) to generate, directly or indirectly, a 3′-fork DNA intermediate suitable to be recognized by the pre-primosomal proteins, an essential step for replication re-start [46]. Here, RecA and its accessory proteins limit further RadA/Sms-mediated unwinding of the 3′-fork DNA.

Finally, RadA/Sms bound to the 5′ tail of HJ-Lag DNA unwinds the nascent lagging-strand to render a 3′-fork DNA substrate upon spontaneous remodeling (Figure 2 and Appendix A). Here, RecA and the SsbA and RecO mediators limit RadA/Sms binding onto and unwinding of the nascent lagging-strand to prevent incorrect unwinding of the template lagging-strand (Figure 2 and Appendix A).

In conclusion, as reported in other systems, in *B. subtilis*, different motor proteins process replicative stress intermediates [48]. However, unlike the fork remodeler RecG or RuvAB, RadA/Sms seems to selectively unwind the nascent lagging-strand of stalled or reversed forks. This way, a branched DNA intermediate that resembles a proper substrate for PriA-dependent replication restart is generated.

## 4. Materials and Methods

### 4.1. Strains and Plasmids

*E. coli* BL21(DE3) (pLysS) cells bearing pCB1020 (*radA*), pCB1035 (*radA* C13A), pCB1038 (*radA* K104R), pCB1037 (*radA* K104A), pCB1040 (*radA* K255R), pCB722 (*ssbA*), or pCB669 (*recO*) encoding the respective gene under the control of a rifampicin-resistant promoter (P_T7_) were used to overproduce their respective proteins [34,42,47,59,60,61]. *B. subtilis* BG214 cells bearing pBT61 (*recA*) were used to overproduce RecA [62].

### 4.2. Enzymes, Reagents, Protein and DNA Purification, Protein–Protein Interaction

All chemicals used were analytical grade. Polyethyleneimine, IPTG (isopropyl-β-D-thiogalactopyranoside), DTT, ATP, ATPγS, and dATP were from Merck (Darmstadt, Germany). DNA ligase, DNA polymerase, and DNA restriction enzymes were from New England Biolabs (Ipswich, MA, USA). DEAE, Q-, and SP-Sepharose were from Cytiva (Marlborough, MA, USA). Hydroxyapatite was from Bio-Rad (Hercules, CA, USA), phosphocellulose was from Whatman (Maidstone, UK), and the Ni-column was from Qiagen (Hilden, Germany).

The protein RadA/Sms and its variants RadA/Sms C13A, RadA/Sms K104R, RadA/Sms K104A and RadA/Sms K255R (49.4 kDa), RecA (38.0 kDa), SsbA (18.7 kDa), and RecO (29.3 kDa) were expressed and purified as described [34,42,47,63]. Purified PcrA (87.3 kDa) was a gift of M. Moreno-del Álamo (this laboratory), and purified RecG (78.1 KDa) was a gift of S. Ayora (this laboratory). The corresponding molar extinction coefficients for RadA/Sms, RecA, SsbA, RecO, RecG, and PcrA were calculated as 24,900; 15,200; 11,400; 19,600; 63,500; and 70,400 M^−1^ cm^−1^, respectively, at 280 nm, as described [64]. Protein concentration was determined using the above molar extinction coefficients. The concentrations of RecA, RecO, RecG, and PcrA are expressed as moles of monomers, SsbA as moles of tetramers, and RadA/Sms (and its mutant variants) as moles of hexamers, unless stated otherwise. In this study, experiments were performed under optimal RecA conditions in buffer A (50 mM Tris-HCl pH 7.5, 1 mM DTT, 80 mM NaCl, 10 mM Mg(CH_3_COO)_2_, 50 μg/mL bovine serum albumin (BSA), and 5% glycerol).

The nucleotide (nt) sequence of the oligonucleotides used is indicated in the 5′→3′polarity: J3-1, CGCAAGCGACAGGAACCTCGAGAAGCTTCCGGTAGCAGCCTGAGCG GTGGTTGAATTCCTCGAGGTTCCTGTCGCTTGCG; J3-2-110, CGCAAGCGACAGGA ACCTCGAGGAATTCAACCACCGCTCAACTCAACTGCAGTCTAGACTCGAGGTTC CTGTCGCTTGCGAAGTCTTTCCGGCATCGATCGTAGCTATTT; J3-2-110-5, AAG TCT TTCCGGCATCGATCGTAGCTATTTCGCAAGCGACAGGAACCTCGAGGAATTCAA CCACCGCTCAACTCAACTGCAGTCTAGACTCGAGGTTCCTGTCGCTTGCG; J3-3, C GCAAGCGACAGGAACCTCGAGTCTAGACTGCAGTTGAGTCCTTGCTAGGACGGA TCCCTCGAGGTTCCTGTCGCTTGCG; J3-4, CGCAAGCGACAGGAACCTCGAGGGA TCCGTCCTAGCAAGGGGCTGCTACCGGAAGCTTCTCGAGGTTCCTGTCGCTTGC G; 170, AGACGCTGCCGAATTCTGGCTTGGATCTGATGCTGTCTAGAGGCCTCCAC TATGAAATCG; 171, CGATTTCATAGTGGAGGCCTCTAGACAGCA; 173, AGCTCAT AGATCGATAGTCTCTAGACAGCATCAGATCCAAGCCAGAATTCGGCAGCGTCT; 172, TGCTGTCTAGAGACTATCGATCTATGAGCT; 171-15, CGATTTCATAGTGGA; 172-15, ATCGATCTATGAGCT; 170-ER, AGACGCTGCCGAGGATCCCGGGTACCAAT CATAGCAATTGCAGCCTCCACTATGAAATCG; 173-ER, AGCTCATAGATCGATAG TGTATTAAGTGAGGATTGGTACCCGGGATCCTCGGCAGCGTCT; 171-ER, CGATTT CATAGTGGAGGCTGCAATTGCTAT; 172-ER, CTCACTTAATACACTATCGATCTAT GAGCT; 173-5, CAGCATCAGATCCAAGCCAGAATTCGGCAGCGTCT; 173-10, CTAG ACAGCATCAGATCCAAGCCAGAATTCGGCAGCGTCT; 173-15, AGTCTCTAGACAG CATCAGATCCAAGCCAGAATTCGGCAGCGTCT; 173-20, TCGATAGTCTTAGACAG CATCAGATCCAAGCCAGAATTCGGCAGCGTCT. In Appendix A, the oligonucleotide composition of the different substrates is described. Annealing was performed by heating the corresponding labelled and cold oligonucleotides in a potassium phosphate buffer for 10 min at 100 °C, followed by slowly cooling them to room temperature. The substrates were gel purified as described [47,59] and stored at 4 °C. In solid lines are denoted the complementary strands, and in dotted lines the noncomplementary regions. The radiolabeled strand is represented in grey color with an asterisk (*) denoting the labelled 5′-end. Half of an arrowhead indicates the 3′-ends. The 3199-bp pGEM3 Zf(+) was used as a source of circular ssDNA and dsDNA (Promega Biotech, Spain). DNA concentrations were established using the molar extinction coefficients of 8780 and 6500 M^−1^ cm^−1^ at 260 nm for ssDNA and dsDNA, respectively, and are expressed as moles of nt or as moles of DNA molecules.

### 4.3. ATP Hydrolysis Assays

The ATP hydrolysis activity of RecA, RadA/Sms (or of its mutant variants), and PcrA was assayed in the presence of RecA mediators via an NAD/NADH coupled spectrophotometric enzymatic assay [47]. Since RadA/Sms and its variants were purified using the same protocol, and the RadA/Sms K104R and RadA/Sms K255R variants show no ATP hydrolysis, we are confident that the measured ATPase activity is associated to the RadA/Sms or RadA/Sms C13A protein.

The rate of ATP hydrolysis was measured in buffer A containing 5 mM ATP and an ATP regeneration system (620 μM NADH, 100 U/mL lactic dehydrogenase, 500 U/mL pyruvate kinase, and 2.5 mM phosphoenolpyruvate) for 30 min at 37 °C [47]. The order of addition of circular 3199-nt pGEM3 Zf(+) (cssDNA) (10 μM in nt) and of the purified proteins (and their concentration) is indicated in the text. Data obtained from A_340_ absorbance were converted to ADP concentrations and plotted as a function of time [47]. Accepting that RecA, *wt* RadA/Sms, or its mutant variant RadA/Sms C13A operates mostly under steady-state conditions, the maximal number of substrate-to-product conversion per unit of time (V_max_) was measured. *t*-tests were applied to analyze the statistical significance of the data. At longer times, ATP hydrolysis seems to decrease, but this is solved by increasing the concentration of the ATP regeneration system, suggesting that at very late times it is rate limiting.

### 4.4. DNA Unwinding Assays

The different forked DNA substrates (0.5 nM) used were incubated with increasing concentrations of RadA/Sms, its mutant variant RadA C13A, or RecG in the presence of the indicated concentrations of RecA, RecO, or SsbA for 15 min at 30 °C in buffer A containing 2 mM ATP in a 20-μL volume. The reactions were deproteinized by phenol-chloroform; DNA substrates and products were precipitated by NaCl and ethanol addition and subsequently separated using 6% (*w*/*v*) PAGE. Gels were run and dried prior to phosphor-imaging analysis. The bands were quantified using ImageJ (NIH). *t*-tests were applied to analyze the statistical significance of the data.

### 4.5. RecA-Mediated DNA Strand Exchange

The 3199 bp *Kpn*I-cleaved pGEM3 Zf(+) dsDNA (20 μM in nt) and the homologous circular 3199-nt ssDNA (10 μM in nt) were pre-incubated with the indicated protein or protein combination in buffer A containing 5 mM ATP or dATP (5 min at 37 °C). Then, a fixed RecA and a fixed or variable *wt* or mutant RadA/Sms concentration were added and the reaction was incubated for fixed or variable times at 37 °C. An ATP regeneration system (8 units/mL creatine phosphokinase and 8 mM phosphocreatine) was included in the recombination reaction. After incubation of the recombination reaction, samples were de-proteinized as described [65] and fractionated through 0.8% agarose gel electrophoresis with ethidium bromide. The signal of the remaining linear dsDNA (*lds*) substrate and the appearance of joint molecule (*jm*) intermediates and nicked circular (*nc*) products were quantified from the gels using a Geldoc system and the Image Lab software (BioRad, Hercules, CA, USA), as described [30]. When indicated, the sum of *jm* and *nc* are shown (% of recombination).

### 4.6. Protein–Protein Interaction Assays

In vitro protein–protein interactions were assayed using affinity chromatography as earlier described [45]. His-tagged RadA/Sms, native SsbA, or RadA/Sms and SsbA (1 µg each) was (were) pre-incubated in buffer A (5 min, 37 °C), both in the presence or absence of 10 μM cssDNA and 5 mM ATP, and then loaded onto a 50 µL Ni^2+^ matrix microcolumn, and the FT was collected. After extensive washing, the retained proteins in the Ni^2+^ matrix were eluted with 50 µL of buffer A containing 400 mM imidazole. The proteins were separated by 10% SDS-PAGE, and gels were stained with Coomassie Blue (Merck, Darmstadt, Germany).

## Figures and Tables

**Figure 1 ijms-24-04536-f001:**
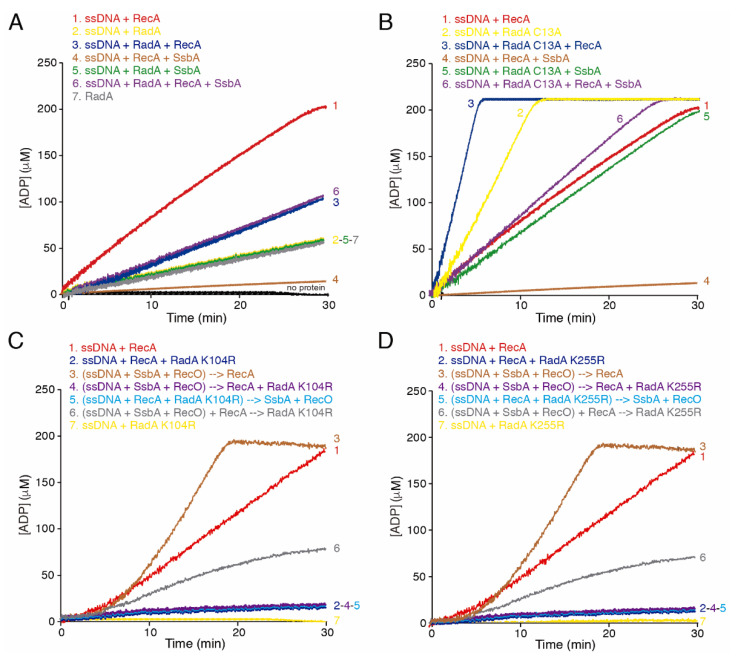
The RecA ATPase is differentially affected by RadA/Sms and the RecO-SsbA mediator. (**A**,**B**) The effect of SsbA on the ATPase activity of RadA/Sms (**A**) or RadA/Sms C13A (**B**) and/or RecA. (**C**,**D**) The effect of the RecO-SsbA mediator on the ATPase activity of RadA/Sms K104R (**C**) or K255R (**D**) and/or RecA. In all reactions, the circular 3199-nt ssDNA (cssDNA, 10 μM in nt) was incubated with the indicated proteins and concentrations (RecA (800 nM), RadA/Sms or its mutant variants (30 nM), SsbA (150 nM), and RecO (100 nM)) in the order detailed, in buffer A containing 5 mM ATP and the ATP regeneration system, and the ATPase activity was measured for 30 min. The arrow denotes that the ssDNA was incubated (5 min at 37 °C) with the preceding protein(s) (denoted between parentheses) to preform cssDNA-protein complex(es). ATP hydrolysis measurement was started after all proteins were added. All reactions were repeated three or more times with similar results, and a representative graph is shown here. The mean ATPase activity V_max_ values ± SEM, calculated from the slope of the curves, can be found in Appendix A.

**Figure 2 ijms-24-04536-f002:**
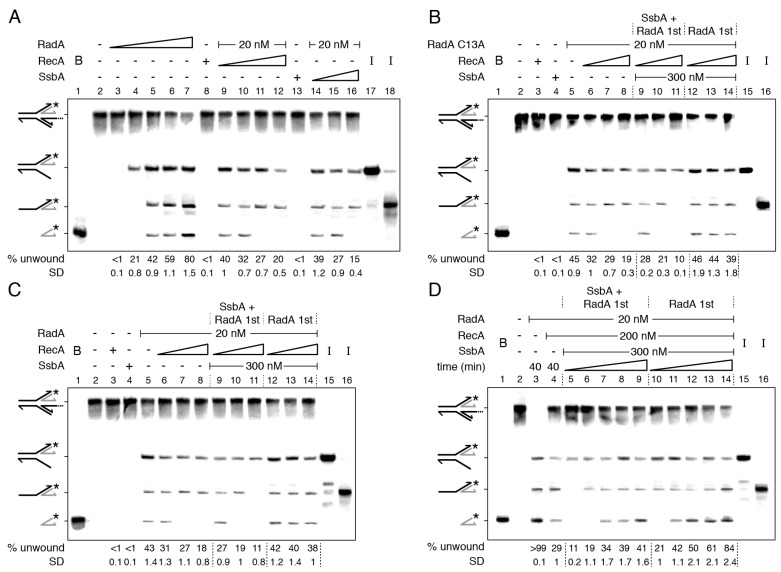
RecA and/or SsbA limit RadA/Sms unwinding of HJ DNA when the nascent lagging-strand is longer than the leading-strand. (**A**–**D**) Left-hand side, illustration of a reversed replication fork with a nascent lagging-strand longer than the leading-strand (HJ-Lag). [γ-^32^P] HJ-Lag DNA (0.5 nM in molecules) was incubated with the indicated proteins (fixed (20 nM) or increasing RadA/Sms (doubling from 5 to 80 nM), RadA/Sms C13A (20 nM), fixed (200 nM) or increasing RecA (doubling from 50 to 400 nM (**A**) or from 100 to 400 nM (**B**,**C**)), and fixed (300 nM) or increasing SsbA (doubling from 75 to 300 nM)) in the order detailed in the figures in buffer A containing 2 mM ATP (15 min at 30 °C) (**A**–**C**) or for a variable time (10, 15, 20, 30, and 40 min at 30 °C) (**D**). When indicated, the DNA was pre-incubated with the indicated protein (s) (5 min at 30 °C) in buffer A, then the second protein (s) and ATP were added. Reactions were stopped, deproteinized, and separated by 6% native PAGE. Gels were dried and visualized by phosphor imaging. The different orders of protein addition were separated by dashed lines. *, denotes the labelled strand; half of an arrowhead, the 3′ends; -, no protein added; B, boiled product; and I, substrate intermediates. The labelled intermediates and product were run in parallel as mobility markers. The fraction of unwound products in three independent experiments, such as those shown in (**A**–**D**), was quantified, and the mean ± SD represented.

**Figure 3 ijms-24-04536-f003:**
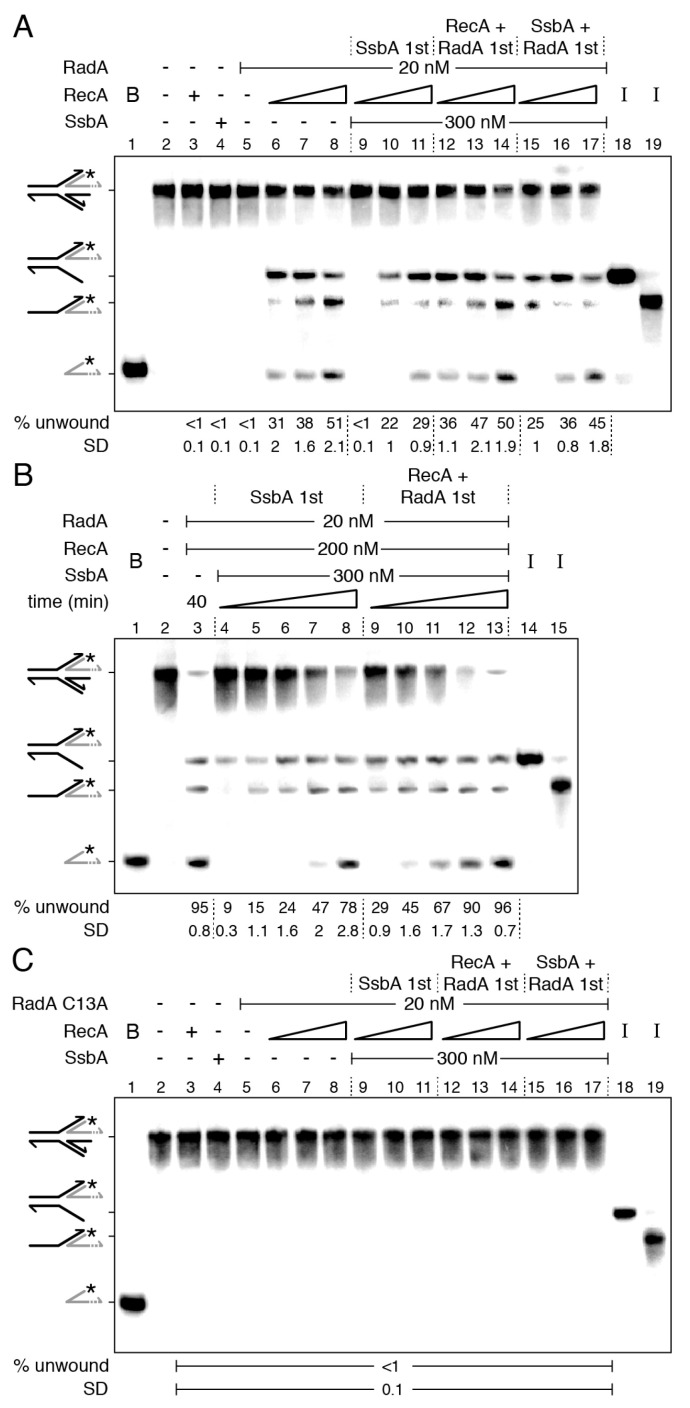
RecA facilitates RadA/Sms unwinding of HJ-Lead DNA when the nascent leading-strand is longer than the lagging-strand. (**A**–**C**) Left-hand side, illustrations of a reversed replication fork with a nascent leading-strand longer than the lagging-strand (HJ-Lead). [γ-^32^P] HJ-Lead DNA (0.5 nM in molecules) was incubated with the indicated protein(s) (fixed RadA/Sms or RadA/Sms C13A (20 nM), fixed (300 nM) or increasing (doubling from 75 to 300 nM) SsbA, and fixed (200 nM) or increasing RecA (100 to 400 nM)) in the order detailed in the figures in buffer A containing 2 mM ATP (for 15 min at 30 °C) (**A**,**C**) or for a variable time (10, 15, 20, 30, and 40 min at 30 °C) (**B**). When indicated, the HJ-Lead DNA was pre-incubated with the indicated protein(s) (5 min at 30 °C) in buffer A, and then the second protein(s) and ATP were added. Reactions were stopped, deproteinized, and separated by 6% native PAGE. Gels were dried and visualized by phosphor imaging. The different orders of protein addition were separated by dashed lines. *, denotes the labelled strand; half of an arrowhead, the 3′ends; -, no protein added; B, boiled product; and I, substrate intermediates. The labelled intermediates and product were run in parallel as mobility markers. The fraction of unwound products in three independent experiments such as those shown in (**A**–**C**) was quantified, and the mean ± SD represented.

**Figure 4 ijms-24-04536-f004:**
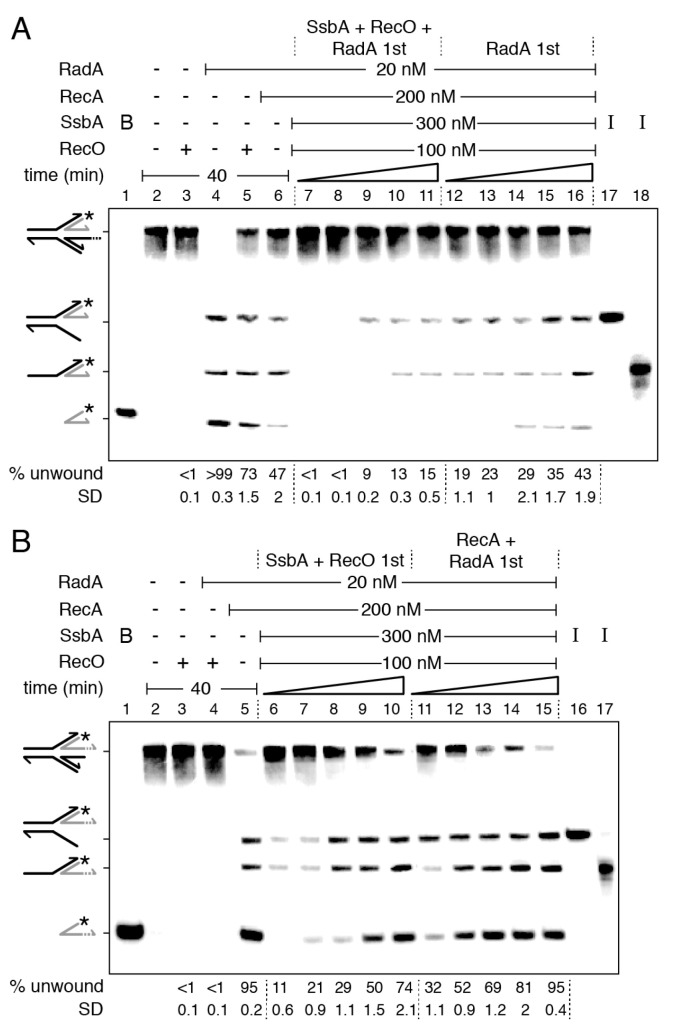
RecO-SsbA limits RadA/Sms unwinding of reversed fork DNA. Left-hand side, illustrations of a reversed replication fork with a nascent lagging- (HJ-Lag [**A**]) or leading-strand (HJ-Lead [**B**]) longer than the complementary strand. (**A**) [γ-^32^P] HJ-Lag or (**B**) [γ-^32^P] HJ-Lead DNA (0.5 nM in molecules) was incubated with the indicated protein(s) (RadA/Sms (20 nM), RecA (200 nM), SsbA (300 nM), and RecO (100 nM)) in the order detailed in the figure in buffer A containing 2 mM ATP for a variable time (10, 15, 20, 30, and 40 min at 30 °C). The HJ DNA substrate was pre-incubated with the proteins denoted by 1st (5 min at 30 °C) in buffer A, then the other protein(s) and ATP were added. Reactions were stopped, deproteinized, and separated by 6% native PAGE. Gels were dried and visualized by phosphor imaging. The different orders of protein addition were separated by dashed lines. *, denotes the labelled strand; half of an arrowhead, the 3′ends; -, no protein added; B, boiled product; and I, substrate intermediates. The labelled intermediates and product were run in parallel as mobility markers. The fraction of unwound products in three independent experiments such as those shown in (**A**,**B**) was quantified, and the mean ± SD represented.

**Figure 5 ijms-24-04536-f005:**
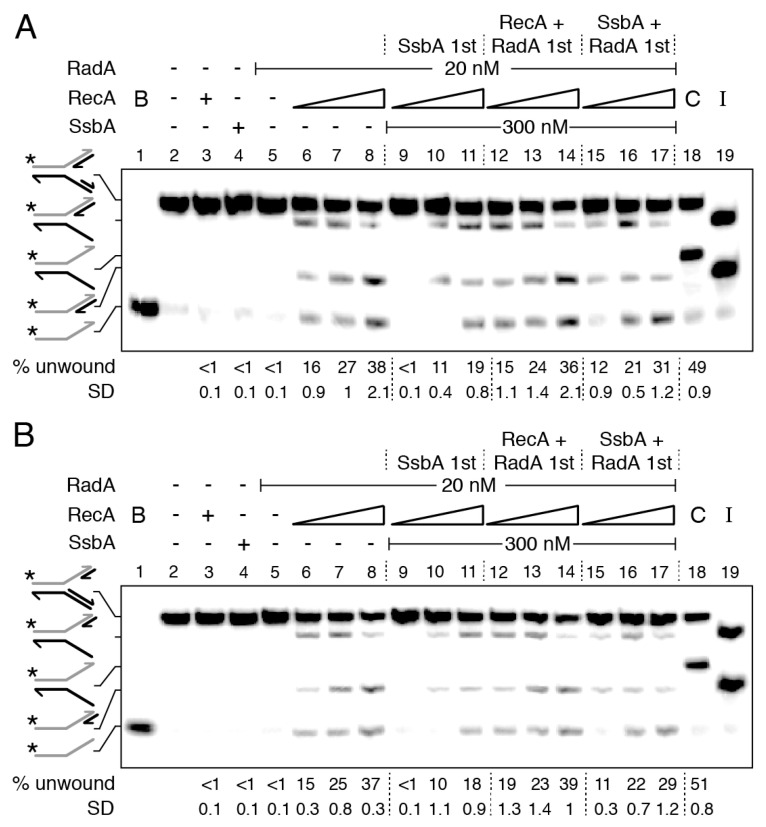
RecA activates RadA/Sms to unwind the nascent lagging-strand of a stalled fork. Left-hand side, illustrations of stalled replication forks with a lagging- (forked-Lag (**A**)) or leading-strand (forked-Lead (**B**)) gap. [γ-^32^P] forked-Lag (**A**) or [γ-^32^P] forked-Lead (**B**) DNA (0.5 nM) was incubated with the indicated protein (s) (RadA/Sms (20 nM), RecA (doubling from 100 to 400 nM), and SsbA (300 nM)) in buffer A containing 2 mM ATP for 15 min at 30 °C in the order detailed in the figure. In some reactions, the forked DNA was pre-incubated with the indicated protein (s) (5 min at 30 °C) in buffer A, then the second protein (s) and ATP were added. Reactions were stopped, deproteinized, and separated by 6% native PAGE. Gels were dried and visualized by phosphor imaging. The different orders of protein addition were separated by dashed lines. *, denotes the labelled strand; half of an arrowhead, the 3′ends; -, no protein added; B, boiled product; C, a control reaction with RecG (15 nM), which branch migrates the DNA substrate, resulting in unreplicated forked DNA; and I, substrate intermediates. The labelled intermediates and product were run in parallel as mobility markers. The fraction of unwound products in three independent experiments such as those shown in (**A**,**B**) was quantified, and the mean ± SD represented.

**Figure 6 ijms-24-04536-f006:**
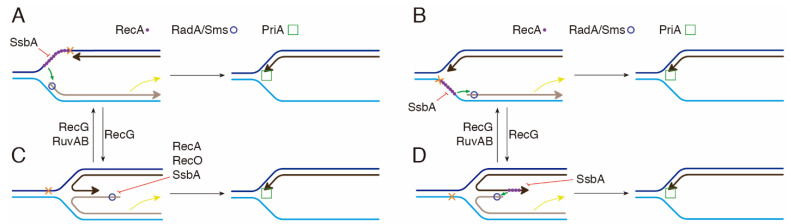
Proposed model for the action of the 5′→3′ RadA/Sms helicase in coordination with RecA and its mediators. The RecG remodeler converts forked-Lead (**A**) or forked-Lag (**B**) onto a HJ-Lag (**C**) or HJ-Lead (**D**) DNA, and RecG or RuvAB restore the reversed fork, leading to fork regression. RecA bound to a stalled fork (**A**,**B**) or a reversed fork with a longer nascent-leading-strand (**D**) recruits RadA/Sms to unwind the nascent lagging-strand to yield a 3′-fork DNA, with SsbA or RecO competing for binding to the ssDNA. RadA/Sms bound on HJ-Lag (**C**) unwinds the nascent lagging-strand to yield a 3′-fork DNA, with RecA, SsbA or RecO competing for binding to the ssDNA. Then, PriA bound to the 3′-fork DNA substrate recruits other pre-primosomal components to reinitiate DNA replication.

## Data Availability

Raw gel images, ATPase data, and materials used in this study are held by the authors and are available on request. The materials used in this study are available from the corresponding author.

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
