# Peer review of "Bacillus subtilis RadA/Sms-Mediated Nascent Lagging-Strand Unwinding at Stalled or Reversed Forks Is a Two-Step Process: RadA/Sms Assists RecA Nucleation, and RecA Loads RadA/Sms"

_ijms, 2023, doi:10.3390/ijms24054536_

Round 1

Reviewer 1 Report

In this manuscript, the authors provide an extensive analysis of the interplay between RadA/Sms, RecA, and mediators SsbA and RecO. The experiments are generally well-designed and the results and their implications are comprehensively discussed. The paper is pretty well-written, albeit a bit dense and at times overly expositive. Prior to publication, there are a few additional experiments that could strengthen the paper that should be undertaken:

First, regarding the effect of SsbA on RadA/Sms ATPase activity, the authors express concern that ssDNA-independent ATPase activity might have clouded their analysis. These experiments were performed at fixed DNA and ATP concentrations. Perhaps examining these combinations of DNA remodeling factors' ATPase activity as a function of ATP concentration or DNA concentration will uncover an effect of SsbA on RadA/Sms ATPase activity or at least eliminate the concern that ssDNA-independent ATPase activity is masking such an effect.

Second, RadA/Sms displacement of SsbA is mentioned a number of times in the manuscript. To most readers, this will imply removal from the DNA in question. In the experimental contexts presented in the manuscript, displacement seems to be kind of an umbrella that covers more potential scenarios than ejection of SsbA from the DNA. Therefore, it would be good to have a control experiment or two that demonstrates whether SsbA is displaced in the traditional sense from DNA by RadA/Sms, perhaps via gel shift combined with (or in parallel to) detection of the displaced SsbA (e.g. by western blotting).

Author Response

Our replies to the reviewers' comments are shown in CAPITAL LETTERS and with a smaller font

Reviewer 1

In this manuscript, the authors provide an extensive analysis of the interplay between RadA/Sms, RecA, and mediators SsbA and RecO. The experiments are generally well-designed and the results and their implications are comprehensively discussed. The paper is pretty well-written, albeit a bit dense and at times overly expositive. Prior to publication, there are a few additional experiments that could strengthen the paper that should be undertaken:

THANK YOU FOR YOUR ENTHUSIASM ABOUT OUR BIOCHEMICAL CHARACTERIZATION OF THE B. SUBTILIS RADA/SMS DNA HELICASE AND ITS INTERPLAY WITH RECA AND ITS MEDIATORS.

First, regarding the effect of SsbA on RadA/Sms ATPase activity, the authors express concern that ssDNA-independent ATPase activity might have clouded their analysis. These experiments were performed at fixed DNA and ATP concentrations. Perhaps examining these combinations of DNA remodeling factors' ATPase activity as a function of ATP concentration or DNA concentration will uncover an effect of SsbA on RadA/Sms ATPase activity or at least eliminate the concern that ssDNA-independent ATPase activity is masking such an effect.

THANK YOU FOR POINTING THAT WE MUST CLARIFY THE MESSAGE ABOUT SSDNA-INDEPENDENT ATPASE ACTIVITY OF RADA/SMS.

IN OUR PREVIOUS PAPERS (TORRES ET AL., DNA REPAIR, 2019; TORRES ET AL., NAR, 2019; TORRES ET AL., FRONT MICROBIOL, 2021) WE HAVE ANALYSED MANY VARIABLES TO UNRAVEL WHETHER THE ATPASE ACTIVITY OF RADA/SMS WAS STIMULATED BY DNA. FOR INSTANCE, WE HAVE VARIED SSDNA OR PROTEIN CONCENTRATIONS, THE METAL ION, THE ATP CONCENTRATION AND THE TEMPERATURE OF INCUBATION. HOWEVER, UNDER NO CONDITION THE STABLE AND REPRODUCIBLE SSDNA-INDEPENDENT ATPASE ACTIVITY OF RADA/SMS WAS STIMULATED BY THE ADITION OF DNA.

OUR CONCERN ABOUT SSDNA-INDEPENDENT ATPASE ACTIVITY OF RADA/SMS OR RADA/SMS C13A WAS AN INTRODUCTION TO EXPLAIN TO READERS WHY WE ARE USING ATPASE MUTANTS TO EVALUATE THE INTERPLAY AMONG THOSE PROTEINS WITH SSBA AND/OR RECA. THE TEXT WAS MODIFIED TO AMELIORATE THE REVIEWER CONCERN.

Second, RadA/Sms displacement of SsbA is mentioned a number of times in the manuscript. To most readers, this will imply removal from the DNA in question. In the experimental contexts presented in the manuscript, displacement seems to be kind of an umbrella that covers more potential scenarios than ejection of SsbA from the DNA. Therefore, it would be good to have a control experiment or two that demonstrates whether SsbA is displaced in the traditional sense from DNA by RadA/Sms, perhaps via gel shift combined with (or in parallel to) detection of the displaced SsbA (e.g. by western blotting).

WE ARE SORRY IF WE HAVE MISLED THE REVIEWER BY USING AN INAPROPRIATE WORD “DISPLACEMENT”.

HOW SSB/SSBA/RPA IS DISPLACED BY POSITIVE RECA MEDIATORS (RECO, BRCA2/RAD52) IS AN OLD AND UNSOLVED DILEMA. RECENTLY, IT HAS BEEN SUGGESTED THAT THE RECO MEDIATOR UPON BINDING DISTORTS THE SSDNA IN A WAY THAT DESTABILIZES SSB BOUND TO SSDNA. THUS, AS THE REVIEWER SUGGESTS, WE AGREE THAT IT WOULD BE INTERESTING TO UNCOVER ALSO THE MECHANISM UNDER THIS RADA/SMS-MEDIATED SSBA DISPLACEMENT. RECENTLY, POLARD AND KOWALCZYKOWSKI LABS HAVE SHOWN, USING CRYOEM STUDIES, THAT STREPTOCOCCUS PNEUMONIAE RADA CAN DISPLACE SSBA AND LOADS RECA ONTO SSDNA (REFERENCE 44).

WE CONSIDER THAT RADA/SMS ACTIVATION BY RECA IS A DYNAMIC EVENT THAT CANNOT BE SEEN WITH STATIC GEL SHIFT EXPERIMENTS. FURTHERMORE, THE DEMOSTRATION THAT RADA/SMS IN CONCERT WITH RECA DISPLACE SSBA, CANNOT BE PERFORMED UNDER THE EXPERIMENTAL CONDITIONS WE USED. WE WERE USING SATURATING SSBA (300 nM) CONCENTRATIONS, AND WE EXPECTED THAT 1 SSBA TETRAMER SHOULD BOUND TO A SSDNA OF THE FORKED/REVERSED MOLECULE, TO MAKE SURE THAT ALL THE DNA IS BOUND BY SSBA. THUS, THE DISPLACEMENT OF SSBA FROM SSDNA COULD NOT BE DETECTED, BECAUSE THERE IS A LARGE EXCESS OF FREE SSBA PROTEIN THAT WOULD BE INDISTINGUISHABLE FROM THAT DISPLACED FROM THE DNA.

WE HAVE SHOWN IN FIGURE 1B THAT RADA/SMS C13A CAN PARTIALLY DISPLACE SSBA FROM SSDNA. IN FIGURE 3, LANE 9, IT IS SHOWN THAT WHEN SSBA IS PREINCUBATED WITH SSDNA, IT INHIBITS RADA/SMS UNWINDING. BY INCREASING THE RECA CONCENTRATION, WHICH BY ITSELF CANNOT DISPLACE SSBA, RADA/SMS IS ACTIVATED TO UNWIND THE NASCENT LAGGING-STRAND AND INDIRECTLY TO DISPLACE SSBA (FIGURE 3, LANE 10-11). THE INHIBITION BY SSBA IS LESS OBVIOUS WHEN SSDNA IS PRE-INCUBATED WITH RADA/SMS AND THEN CHALLENGED WITH SSBA. THE TEXT WAS MODIFIED TO ANSWER THE QUERY

Reviewer 2 Report

In B. subtilis, the replication machinery undergoes pervasive cycles of disassembly and reassembly. Moreover, upon encountering replication obstacles, cells form RecA foci that colocalize with replisomes. In this context, RecA does not trigger the SOS response. Instead, it is thought that RecA assists in replication fork remodeling and recovery. This includes recruiting the loader of the replicative helicase DnaD and the replicative polymerase PolC at the site of stalled replication forks. Torres et al. expand on this process further with this elegant biochemical reconstitution study. The authors present in vitro evidence that suggests that RecA and its regulator RadA/Sms function in a two-step reaction to first allow RecA nucleation at stalled forks, prior to loading of RadA/Sms in an apparently RecA-assisted manner. Interestingly, this requires direct protein-protein interactions between RecA and RadA, suggesting an active mechanism with implications for the type of reactions that may be possible at stalled forks to mediate recovery. While the model in Fig. 6 remains speculative as it is based purely on in vitro results, it is nonetheless plausible and placed very well in the context of published genetic and cell-biological work, making the mechanistic results presented in the current MS conceptually interesting, with potential implications beyond B. subtilis.

The biochemistry is polished with good controls, albeit slightly complicated to follow. The authors might consider re-naming some of the DNA substrates to help reader. For example, line 100: "5´-tailed 3´-fork DNA (a replicating fork with a fully synthesized leading-strand and no-synthesis in the lagging-strand)". It is quite difficult to understand where and why some of these substrates were deployed and, moreover, constant referral to the substrate table is required, perhaps this could be avoided by better specifying substrates in the text and figures.

While the conclusions are necessarily speculative as to the biological significance of the observed biochemical reactions, the authors are careful in pointing out caveats, alternative explanations, and do not over-interpret the data. The biochemistry is plausibly linked to published in vivo observations.

Minor points:

-In the order of addition experiments, e.g., Fig. 2, 3, 4, 5, it should be clearly indicated within which lanes a certain regimen is used and where this changes.

-Fig. 4S is not easy to read. For example, it is not spelled out that ‘jm’ stands for ‘joint molecules’. A representative cartoon (for example, Carrasco et al. 2022, by the authors) would help.

-Please review the use of "compete" (with/against/outcompete....), e.g., line 325: "RecA and SsbA cannot compete a preformed RadA/Sms-HJ-Lag DNA complex".

Author Response

Our replies to the reviewers' comments are shown in CAPITAL LETTERS and with a smaller font

Reviewer 2

In B. subtilis, the replication machinery undergoes pervasive cycles of disassembly and reassembly. Moreover, upon encountering replication obstacles, cells form RecA foci that colocalize with replisomes. In this context, RecA does not trigger the SOS response. Instead, it is thought that RecA assists in replication fork remodeling and recovery. This includes recruiting the loader of the replicative helicase DnaD and the replicative polymerase PolC at the site of stalled replication forks. Torres et al. expand on this process further with this elegant biochemical reconstitution study. The authors present in vitro evidence that suggests that RecA and its regulator RadA/Sms function in a two-step reaction to first allow RecA nucleation at stalled forks, prior to loading of RadA/Sms in an apparently RecA-assisted manner. Interestingly, this requires direct protein-protein interactions between RecA and RadA, suggesting an active mechanism with implications for the type of reactions that may be possible at stalled forks to mediate recovery. While the model in Fig. 6 remains speculative as it is based purely on in vitro results, it is nonetheless plausible and placed very well in the context of published genetic and cell-biological work, making the mechanistic results presented in the current MS conceptually interesting, with potential implications beyond B. subtilis.

WE WOULD LIKE TO THANK THE REVIEWER FOR HER/HIS ENTHUSIASM ABOUT OUR BIOCHEMICAL CHARACTERIZATION OF THE B. SUBTILIS RADA/SMS DNA HELICASE AND ITS INTERPLAY WITH RECA AND ITS MEDIATORS. THESE RESULTS OPEN THE WAY TO CHALLENGE OUR MODEL BY GENETIC AND CYTOLOGICAL MEANS.

The biochemistry is polished with good controls, albeit slightly complicated to follow. The authors might consider re-naming some of the DNA substrates to help reader. For example, line 100: "5´-tailed 3´-fork DNA (a replicating fork with a fully synthesized leading-strand and no-synthesis in the lagging-strand)". It is quite difficult to understand where and why some of these substrates were deployed and, moreover, constant referral to the substrate table is required, perhaps this could be avoided by better specifying substrates in the text and figures.

OUR APOLOGIES. THE TEXT WAS REPHRASED TO ANSWER THE QUERY.

While the conclusions are necessarily speculative as to the biological significance of the observed biochemical reactions, the authors are careful in pointing out caveats, alternative explanations, and do not over-interpret the data. The biochemistry is plausibly linked to published in vivo observations.

WE WOULD LIKE TO THANK AGAIN THE REVIEWER FOR HER/HIS COMMENTS ABOUT OUR BIOCHEMICAL STUDIES.

Minor points:

-In the order of addition experiments, e.g., Fig. 2, 3, 4, 5, it should be clearly indicated within which lanes a certain regimen is used and where this changes.

THE ORDER OF ADDITION WAS CLARIFIED IN THE FIGURES.

-Fig. 4S is not easy to read. For example, it is not spelled out that ‘jm’ stands for ‘joint molecules’. A representative cartoon (for example, Carrasco et al. 2022, by the authors) would help.

THANK YOU FOR THE COMMENT, FIG S4 WAS MODIFIED.

-Please review the use of "compete" (with/against/outcompete....), e.g., line 325: "RecA and SsbA cannot compete a preformed RadA/Sms-HJ-Lag DNA complex".

THANK YOU, THE TEXT WAS MODIFIED.

Round 2

Reviewer 1 Report

The authors have satisfactorily addressed the issues I raised. In my opinion the manuscript is acceptable for publication.